# Recrystallization of Si Nanoparticles in Presence of Chalcogens: Improved Electrical and Optical Properties

**DOI:** 10.3390/ma15248842

**Published:** 2022-12-11

**Authors:** Alexander Vinokurov, Vadim Popelensky, Sergei Bubenov, Nikolay Kononov, Kirill Cherednichenko, Tatyana Kuznetsova, Sergey Dorofeev

**Affiliations:** 1Department of Chemistry, Lomonosov Moscow State University, Leninskie Gory, 1–3, 119991 Moscow, Russia; 2Prokhorov General Physics Institute of the Russian Academy of Sciences, St. Vavilov, 38, 119991 Moscow, Russia; 3Department of Colloidal and Physical Chemistry, Gubkin Russian State University of Oil and Gas, Leninsky Avenue, 65, 119991 Moscow, Russia

**Keywords:** nanosilicon, chalcogens, doping, semiconductor, sulfur, selenium, tellurium, conductivity

## Abstract

Nanocrystals of Si doped with S, Se and Te were synthesized by annealing them in chalcogen vapors in a vacuum at a high temperature range from 800 to 850 °C. The influence of the dopant on the structure and morphology of the particles and their optical and electrical properties was studied. In the case of all three chalcogens, the recrystallization of Si was observed, and XRD peaks characteristic of noncubic Si phases were found by means of electronic diffraction for Si doped with S and Se. Moreover, in presence of S and Te, crystalline rods with six-sided and four-sided cross-sections, respectively, were formed, their length reaching hundreds of μm. Samples with sulfur and selenium showed high conductivity compared to the undoped material.

## 1. Introduction

Silicon is the second-most abundant element on Earth, and, at the same time, the main semiconductor used for microelectronics. Silicon is also used to create solar panels, biosensors and thermoelectric and other devices. However, the scope of silicon applications is, generally, limited by the properties of its cubic form (also known as Si-I), which is an indirect-gap semiconductor [1]. It is known that the intensity of indirect transitions is much lower than that of direct ones, which renders them unusable for light-emitting devices and adversely affects their photodetector and photovoltaic performance.

This problem can be solved by switching to other silicon allotropes, many of which possess intriguing optical and electrical properties [2,3,4,5], including direct and quasidirect transitions [6,7,8,9]. For example, hexagonal Si-IV was shown to exhibit intense luminescence on the edge of the visible spectrum, with an energy of 1.5 eV [10]. However, the synthesis of noncubic silicon forms is a complicated task, because most of them are not stable and are prone to transitioning back to cubic silicon. At pressures above 12 GPa, Si-I transforms into Si-II with a β-Sn structure [11,12], but after the pressure is released at room temperature, it spontaneously transitions into Si-III. However, Si-III transforms into Si-IV only upon heating, and the latter transitions into Si-I only when annealed. Thus, Si-IV and Si-III have been considered possible to stabilize at ambient conditions [13]; moreover, there are multiple polytypes for Si-IV, which differ in the pattern of layer packing (AB, ABCB, ABCACB, called 2H-Si, 4H-Si and 6H-Si, respectively). Their existence was first predicted with calculations [7,9,14] and later proven experimentally [15].

Therefore, if it were possible to synthesize Si-IV at temperatures below the point of the Si IV → Si I transition, an effective and accessible way to obtain the direct-gap form of silicon could be discovered, which, in turn, has great potential for the production of electronic and optoelectronic devices [4]. As of now, Si-IV can be synthesized using a few methods, including applying extreme pressures [16,17,18,19,20], a structure transfer from a hexagonal GaP [21,22,23], the laser irradiation of Si nanowires [24] and the vapor–liquid–solid (VLS) synthesis of nanowires, which results in a phase mixture [25,26]. The formation of the hexagonal phase is kinetically driven [27]. Sometimes, the hexagonal phase is simultaneously doped with group III and V elements during the VLS procedure [28,29].

As shown in [15,25], the presence of hexagonal Si in nanowires is hard to determine, because diffraction patterns and HRTEM images obtained from hexagonal Si can coincide with those obtained from cubic Si in certain axes, such as <111>, or due to defects in the structure of cubic silicon, especially since hexagonal silicon can revert to its cubic form during synthesis [5]. Moreover, hexagonal and defective cubic Si can exhibit a similar Raman shift. For unequivocal proof, HAADF imaging in STEM can be used [25].

Previously, we obtained silicon microrods in the presence of sulfur vapor. The properties of these rods indicated the formation of noncubic silicon [30]. A feature of this work is the absence of a metal catalyst. Some metals, for example, nickel, facilitate the crystallization of amorphous silicon. Nickel’s role is in the formation of the octahedral crystallite (“nodule”) of the cubic phase of NiSi_2_ with a lattice parameter very close to that of cubic silicon. Such crystallites initiate the crystallization of amorphous nanosilicon (nc-Si) on {111} facets. After that, the NiSi_2_ phase is decomposed due to the replacement of Ni atoms with Si with the simultaneous formation of new NiSi_2_ on the interface between the crystallite and amorphous silicon. Therefore, the NiSi_2_ crystallite migrates through the amorphous silicon phase due to the diffusion of Ni atoms, which results in the crystallization of silicon along the <111> direction [31]. In the case of sulfur, the formation of microrods [30] can be a result of the crystallization of the amorphous part of the nc-Si sample during the formation of the orthorhombic SiS_2_ phase (PDF 3–144; ***a*** = 5.60 Å; ***b*** = 5.53 Å; ***c*** = 9.55 Å; T_melt_ ~ 1100 °C) [32], parameters ***a*** and ***b*** of which are close to that of cubic silicon (***a*** = 5.47 Å). The SiS_2_ phase may work here as a crystallization inductor, similarly to NiSi_2_.

Systems where elements of groups III and V are used for the doping of silicon and obtaining other silicon structures are well studied, but there is currently a limited number of works investigating the interaction of silicon with elements of group VI [33], especially in the context of obtaining noncubic forms of silicon. The objectives of this work were to determine the optimal experimental conditions for the synthesis of nanosilicon microrods with sulfur doping, and to study the effect of heavier sulfur analogs—selenium and tellurium—on the structure and electrophysical properties of nc-Si after doping. These chalcogens were chosen because selenium, like sulfur, forms the orthorhombic phase SiSe_2_ (PDF 25–756; ***a*** = 9.68 Å; ***b*** = 6.00 Å; ***c*** = 5.81 Å; T_melt_ = 972 °C) [34], which can also be an inductor for amorphous silicon crystallization. In the case of tellurium, crystallization from melt is possible when the Si(solid)–melt equilibrium is established at temperatures above 683 °C [35].

## 2. Materials and Methods

Si nanoparticles (nc-Si) were synthesized with the laser pyrolysis of SiH_4_, as described in [36]. The following reagents were used during this work: sulfur, selenium, tellurium, concentrated HNO_3_ (68%), concentrated HF (50%). All the chemicals were purchased from Sigma Tec LLC, and were of reagent grade. The doping of nc-Si was conducted with the diffusion of chalcogen from the gas phase. Samples of sulfur and 10 mg of nc-Si were put in quartz ampoules, which were evacuated, sealed and then annealed in the muffle furnace SNOL-3/11 (“Technoterm”).

Synthesized samples were dispersed in water in an ultrasonic bath (Sono Swiss SW3H) and boiled with an equal volume of concentrated HNO_3_ for 5 min to remove the residual chalcogen. After this, the solution was centrifuged for 1 min at 18,000 rpm, the supernatant was drained, the sample was dispersed in 2 mL of water and then centrifuged again to remove traces of acid. The cleaning procedure was repeated three times. HF etching was performed through the addition of an equal volume of concentrated HF to the solution and sonication for 1 min. Afterwards, the precipitate was washed with water three times, as described above.

A composition analysis was conducted using a Total X-ray Fluorescence (TXRF) spectrometer Picofox S2 (Bruker) with total internal reflection; the samples were in the form of films on sapphire substrates. Scanning electron microscopy (SEM) images were acquired with a SUPRA 40 (Carl Zeiss AG) microscope. For transmission electron microscopy (TEM) and electron diffraction (ED), a JEM-2100 (JEOL) microscope was used. Absorption spectra were recorded on a Cary 50 (Varian) spectrometer. Ultra-high vacuum (UHV, 3 × 10^−9^ torr) measurements were conducted inside the chamber of an Auger spectrometer JAMP-10 (JEOL). The conductivity of samples was measured with a potentiostat P8-nano (Elins). The electrical properties of etched nc-Si sols were investigated on microhotplates that are normally used for gas sensors, as discussed elsewhere [30]. Before electrical measurements, the films were annealed in UHV at 900 °C for 6 min in the case of Si:S and Si:Te, and at 700 °C for 42 s in the case of Si:Se. The registration of the current–voltage characteristics was carried out in the voltage range −15–+15 V at a scanning rate of 300 mV/s.

## 3. Results and Discussion

### 3.1. Annealing of Nanosilicon with Sulfur

To determine the optimal conditions for obtaining silicon microrods, a series of annealings in the presence of sulfur (20 at.%) at varied times and at temperatures in the range of 800–850 °C was carried out. Synthetic conditions and results from our previous work [30] are also included in Table 1 for comparison (see samples Si:S500, Si:S700 and Si:S900).

From Table 1, it follows that the minimum annealing temperature at which the formation of extended microrods was observed was 825 °C. The sulfur content in the purified, but not etched, samples weakly correlated with the amount of sulfur used for the synthesis, and varied in the range of several atomic percent. After etching off the surface oxide layer, the concentration of sulfur dropped by an order of magnitude, and amounted to tenths of a percent, which still significantly exceeded the solubility of sulfur in crystalline silicon (~2 × 10^−4^ at.%) [37]. Such a high sulfur content may have been due to the admixture of the SiS_2_ phase that formed during the annealing phase and became the inclusion inside of the microrods. Another explanation could be the nonequilibrium conditions of recrystallization, which resulted in a supersaturated solution of sulfur in the silicon.

Increasing the temperature and duration of the annealing led to a greater number of observed microrods, as illustrated by Figure 1A–C. After 1 h of annealing at 850 °C, rare hexagonal microrods with a diameter of 5 μm were observed. Longer annealing resulted in an increased number of microrods, their diameter varying in the 5–20 μm range; moreover, objects with a wavy surface were observed, which could have been the result of crystallization in other crystallographic directions. In all cases, the lengths of the microrods varied from dozens to hundreds of μm.

### 3.2. Annealing of Nanosilicon with Selenium and Tellurium

Based on optimal conditions for annealing with sulfur, initial conditions for annealing silicon with selenium and tellurium were chosen—20 at.% of dopant, at 850 °C and for a 5 h duration. These conditions were used as a starting point, and were optimized later on. In Table 2, the results of the annealings with all three chalcogens are presented.

As it was stated above, annealing with sulfur led to the formation of hexahedral rods that, supposedly, had hexagonal lattices [30]. Annealing with selenium did not result in the formation of rods, but faceted crystallites 0.2–1 μm in size were observed (Figure 2), resembling truncated octahedrons in shape. In the case of tellurium, both faceted crystallites and large tetrahedral rods 30–50 μm long and 2–5 μm wide (Figure 3A) were present, as well as druses (Figure 3B). The presence of these crystalline formations in selenium- and tellurium-containing samples seemed to be a consequence of the increased energy favorability of the {100} facets of silicon in the presence of chalcogen vapors. In the case of Te, these facets were shown to have the lowest free surface energy [38].

The sulfur and selenium concentrations in the samples decreased slightly after etching with HF (Table 2), which was a result of the removal of the oxide layer that contained adsorbed chalcogens or silicon chalcogenides. In the case of tellurium, the concentration stayed the same. In all cases, the concentration of chalcogen after etching exceeded its equilibrium solubility in silicon, which was ~2 × 10^−4^ at.% [37,39,40]. For both sulfur and selenium, the high impurity content could be explained in a similar manner: the process of their crystallization from the amorphous phase involves silicon dichalcogenides, which are trapped inside the forming crystals (see Section 3.1). The Si–Te system differs significantly, as there are two phases that may form—Si_2_Te_3_ and SiTe_2_—both of which undergo peritectic decomposition at 684 and 441 °C, respectively [35]. At the annealing temperature of 850 °C, the only possible equilibrium was of Si (solid)–melt (65 at.% Te); therefore, the formation of the faceted structures was only possible through the crystallization from melt. The constant concentration of tellurium of 0.6 at.% before and after etching was, apparently, caused by the solid solution—a high tellurium content in the growth droplets contributed to the average tellurium content in the samples.

### 3.3. Electron Diffraction

Four samples were chosen for the study of the crystal structures, three of which (Si:S5, Si:Se and Si:Te) were synthesized under the same conditions. A lower temperature for sample Si:S2 was chosen for comparison. The crystal structure was examined with a selected area ED during the transmission electron microscopy. For comparison purposes, ICSD and ICDD database entries were used for 2H-Si and cubic Si, respectively; X-ray diffraction data for 4H-Si were taken from a different source [15]. Isolated faceted crystallites were chosen for the measurements. The whole range of the obtained results could be understood from a comparison of the interplanar distances between Si:Te and Si:S5, each sample representing an extreme case (Figure 4). For the Si:Te sample, all interplanar distances corresponded to the cubic phase only; moreover, electron diffraction patterns were perfectly indexed in accordance with such structures (Figure 5C). The Si:S5 sample possessed a wider variety of interplanar distances, including “satellites” of the cubic phase peak at 3.14 Å, which are characteristic of hexagonal phases. Diffraction patterns for most of the crystallites in the Si:S5 sample (Figure 5A) were indexed in accordance with one of the hexagonal polytypes (2H-Si, 4H-Si and 6H-Si). In Si:Se and Si:S2, most of the studied crystallites belonged to the cubic phase; however, in some cases, diffraction patterns could only be explained with the hexagonal structure of 4H-Si (Figure 5B).

The result for Si:S5 coincided with what we observed in [30] for a sample obtained under almost identical conditions. In the case of the Si:S2 sample, a synthetic temperature was probably too low for an effective recrystallization process. The absence of a perfect match with the 2H-Si data could be explained by the decomposition of the hexagonal phase right after it was formed due to its metastability. The minimum temperature for the formation of the rods was 825 °C, which was higher than the decomposition temperature of 2H-Si from the literature [31].

### 3.4. Optical Absorption

The absorption spectra of the Si:S2, Si:S5, Si:Se and Si:Te samples are shown in Figure 6. To exclude the influence of light scattering during the registration of the absorption spectra, they were recorded at four different concentrations—the initial one and diluted two-, three- and four-fold. The absorption spectra of the diluted Si:S2 sample is shown in Appendix A of the Appendix A. This procedure proved that, for all samples, the spectral shape did not depend on concentration (in the studied range of concentrations). Selenium- and sulfur-containing samples (but not the tellurium one) exhibited pronounced absorption bands, with a maximum of 1.8–2.8 eV. A possible explanation of such a phenomenon is the absorption of free carriers.

To prove this assumption, the absorption spectra of the Si:S2, Si:S5 and Si:Se samples were approximated in the energy range of 1–4 eV using functions defined using Drude’s absorption, i.e., the following function: $\alpha \left(\omega \right)=\frac{2\omega k}{c}=\frac{\omega_{p}^{2}\tau }{c\cdot n\cdot \left(1+{\left(\omega -\omega_{0}\right)}^{2}\tau^{2}\right)}=\frac{e^{2}N_{e}\tau }{m^{*}\varepsilon_{0}c\;n\left(\omega \right)}\cdot \frac{1}{\left(1+{\left(\omega -\omega_{0}\right)}^{2}\tau^{2}\right)}$, which was centered at the resonance angular frequency ω_0_, determined the maximum of the Drude contour. In the given function: $\omega_{p}^{2}=\frac{e^{2}N_{e}}{m^{*}\varepsilon_{0}}$–—the plasmon angular frequency; N_e_—the electron concentration; ε_0_—the dielectric constant of the vacuum; n(ω)—the refractive index; τ—a quantity that determines the characteristic time of the interaction of an electron with the electromagnetic field. Fits for the Si:Se, Si:S2 and Si:S5 samples are shown in Figure 7. From these fitting functions, values of ω_0_, τ and N_e_ were determined (Table 3).

Despite the fact that the shape of these absorption bands was successfully described in terms of the Drude theory, the calculated free-electron concentrations were too high—they were several times higher than the concentration of sulfur and selenium in these samples. An increased sub-band-gap absorption was described in the literature for very closely related materials—mono- and nanocrystals of supersaturated solid solutions of chalcogens in cubic silicon, which were obtained as a result of ion implantation, followed by laser annealing. The spectral range in which these materials exhibited excessive absorption compared to pure silicon extended from 0.5 to 3.1 eV [41,42,43], and the content of chalcogen impurity in this case reached 1 at.%. The maximum of this additional absorption band laid in the infrared range: in the literature, the energy of the maximum did not exceed 1.2 eV, but in the present work, this maximum laid at 2.5–2.6 eV. The authors of [41] attributed this sub-band-gap absorption to the formation of a miniband from the levels of impurity defects. In silicon crystals, sulfur and selenium gave rise to a multitude of deep impurity levels corresponding to single defects, dimers and clusters [44]. These levels were located 0.08–0.6 eV lower than the bottom of the conduction band; therefore, for a direct transition from the emergent miniband to Γ-point of the conduction band, one could expect an energy of 2.28–2.8 eV, which was in good agreement with the observed positions of the maxima. The absorbance maximum of ~2 eV was also predicted for a supersaturated (0.4 at.%) selenium solution in silicon by using quantum mechanical calculations [45]. The shift in the optical absorption maximum toward the visible region could be a result of either a higher impurity concentration compared to the literature or a different nature of defects originating from our experimental procedures.

Samples of chalcogen-hyperdoped silicon with pronounced sub-band-gap absorption had a charge carrier concentration at the level of 10^19^ cm^−3^ [46]; low-temperature measurements and quantum chemical calculations indicated the metallic nature of their electronic conductivity [45]. The insulator–metal transition in the case of sulfur happened at concentrations from 0.36 to 0.86 at.% [46], and in the case of selenium, at approximately 0.4 at.%. The obtained concentration of selenium exceeded this boundary value 1.5-fold, while in the case of sulfur, the dopant concentration was less than, or nearly equal to, the limit value. Therefore, we could assume that only the Se-doped sample was degenerate in our case.

### 3.5. Electrical Properties of Films

Conductivity measurements were carried out for the solution-processed films of the Si:S5, Si:Se and Si:Te samples, for which, according to our conclusions (see Section 3.4), an insulator–metal transition was observed. Before the measurements, all films were annealed in UHV; this treatment did not significantly alter the size and morphology of the particles according to SEM.

#### 3.5.1. Si:S5 Film

The I–V characteristics for the Si:S5 film are shown in Figure 8. The shape of the curves indicated that the I–V characteristics of this sample were asymmetric with regard to the forward and reverse currents. The main feature of these I–V characteristics was a strong dependence on forward current strength upon exposure to air. At the same time, the strength of the reverse current did not show this dependency for the first hour of exposure (see curves 1–3, Figure 8). A significant increase in the reverse current could be observed over longer time intervals, while the magnitude of the forward current practically did not change (Figure 8, curve four).

The analytic approximations given in Appendix A of the Appendix A showed that the forward current in the vacuum at voltages below 0.4 V was well described using Ohm’s law, and was determined using the Shockley formula: I_forw_ = I_01_∙V^1.16^ + I_02_(e^V/A^ − 1) at higher voltages. The second term in this formula corresponds to the existence of an energy barrier, which may be a Shottky-type barrier between the sample and the measuring electrode or between the particles. reverse current in the vacuum was approximated with the following function: I_rev_ = I_01_∙V^1.5^ + I_02_∙(V − V_0_)^5^∙exp(−(V − α)/β). The first term of this function defines the charge transfer in the presence of shallow traps (ballistic mode), and the second term defines the impact of the tunneling current [47].

Exposure to air for a prolonged time period (>12 min) changed the Shockley-type approximation function to I_forw_ = I_01_∙V^1.5^ + I_02_∙V^5.5^∙exp(−V/η), similar to the reverse current in the vacuum. The reverse current, upon exposure to air, was approximated with the following function: I_rev_ = I_01_∙V^1.5^ + I_02_∙V^2^ + I_03_·(V − V_0_)^5^·exp(−V/θ). In this function, the first two terms are related to the charge transfer in the presence of shallow traps, and the third term is related to tunneling.

#### 3.5.2. Si:Se Film

The I–V characteristics for the Si:Se film, either in the vacuum or after exposure to air (see Figure 9), were symmetrical and defined using Ohm’s law (approximations are provided in Appendix A in the Appendix A). The absence of barriers and limitations during the charge transfer performed by the space charge in such a highly dispersed system may have indicated a metallic nature of the conductivity, i.e., semiconductor degeneracy.

#### 3.5.3. Si:Te Film

The I–V characteristics for the Si:Te film are shown in Figure 10. The shape of the curves showed a similarity to the Si:S5 film, but the degree of asymmetry for the forward and reverse currents was lower. Same as for the Si:S5 film, the current’s strength decreased significantly after exposure to air.

Analytic approximations (provided in Appendix A) of the I–V characteristics of this sample in the vacuum were given with the three-term function I = I_01_∙V + I_02_∙V^1.5^ + I_03_∙V^α^, where α equals 3.2 for the forward current and 3.5 for the reverse current. Tunneling and barrier transfers gave minimal to no contribution to the current dependences in the vacuum, as there were no exponential terms in the approximation function. On the other hand, exposure to air led to a forward current function transformation into an exponential one, I_forw_ = I_01_∙V^0.5^∙exp(V/V_01_), which was the consequence of oxidation and the concomitant formation of barriers. In the meantime, the reverse current function remained nonexponential: I_rev_ = I_01_∙V^1.5^ + I_02_∙V^2.7^.

Electrophysical measurements largely corroborated previous findings of UV–Vis absorption spectroscopy and TXRF; the degeneracy of the Si:Se sample’s NCs translated into the metallic behavior of the whole composite (film). The insulator–metal transition in the nanoscale disperse systems obeyed a stricter criterion with respect to the bulk material [48], as the contact area between the particles and the presence of insulator layers had to be taken into account. The respective phenomena of hopping and tunneling appeared to have a significant impact on the conductivity of sulfur- and tellurium-doped samples with an insufficient impurity content.

## 4. Conclusions

The annealing of nanosilicon in S, Se and Te vapors at 850 °C led to the recrystallization of compact faceted crystallites one to two orders of magnitude larger than the initial nanoparticles. The crystalline structure of the formed particles varied with the nature of chalcogen, from purely cubic for tellurium to mixed for selenium to predominantly hexagonal (2H, 4H, 6H) for sulfur, as evidenced by using the SAED. Faceted rods were formed for the S and Te samples, tens to hundreds μm in length and 2–20 μm in diameter. Annealing with sulfur led to hexagonal section rods, with tellurium, it led to square section ones, while selenium samples did not contain any rods. The minimal temperature for rod formation was 825 °C; annealing with sulfur at 850 °C for 5 h yielded hexagonal-shaped rods as the main product. Upon the purification of the samples, with the goal of the removal of any residual silicon chalcogenides or elemental chalcogens, the dopant concentration was found to be in the range of 0.12–0.64 at.%, the value exceeding the bulk solubility by three orders of magnitude. The absorption spectra for the sulfur and selenium samples revealed broad bands with maxima in the range of 1.8–2.8 eV, which we associated with the formation of an impurity sub-band and/or with a high concentration of charge carriers. The last conclusion was substantiated due to the increased conductivity of sulfur and selenium solution-processed films after annealing in a high vacuum. The conductivity of the Si:Te film was on par with that of the undoped material, which may have been due to most of the tellurium in the particles not being electrically active. Exposure to air impaired conductivity in all cases. The observed electrophysical properties of the chalcogen-doped nc-Si samples may lead to the creation of sensors; their unusual absorption characteristics are beneficial for photovoltaics. Silicon microrods, on the other hand, may be of use in microelectronic and micromechanical devices.

## Figures and Tables

**Figure 1 materials-15-08842-f001:**
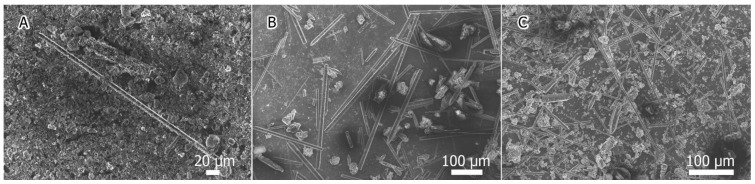
Microphotographs of three nc-Si samples with 20 at.% of sulfur, annealed at 850 °C for 1 (**A**), 3 (**B**) and 5 (**C**) h, respectively.

**Figure 2 materials-15-08842-f002:**
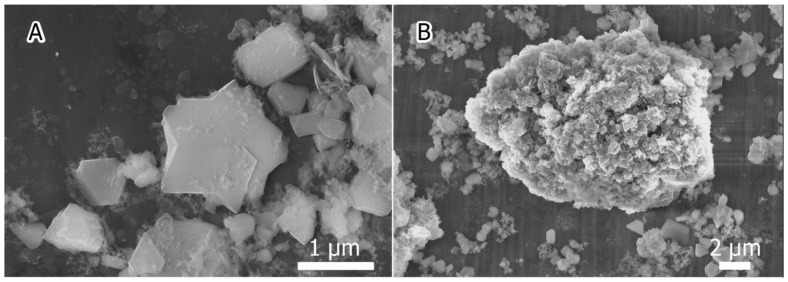
Faceted crystallites observed in Si:Se sample: (**A**) truncated octahedrons; (**B**) recrystallized nanosilicon.

**Figure 3 materials-15-08842-f003:**
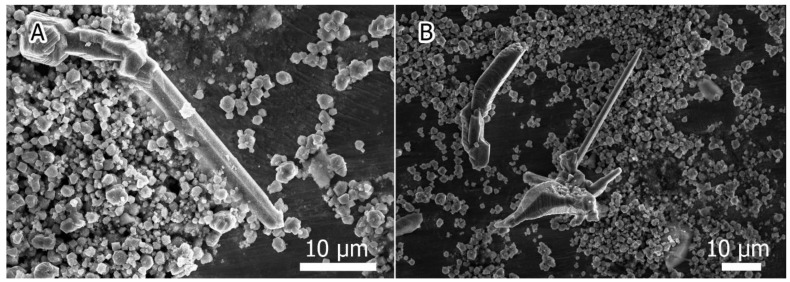
Faceted crystallites and microrods observed in Si:Te sample: (**A**) large tetrahedral rod; (**B**) druses.

**Figure 4 materials-15-08842-f004:**
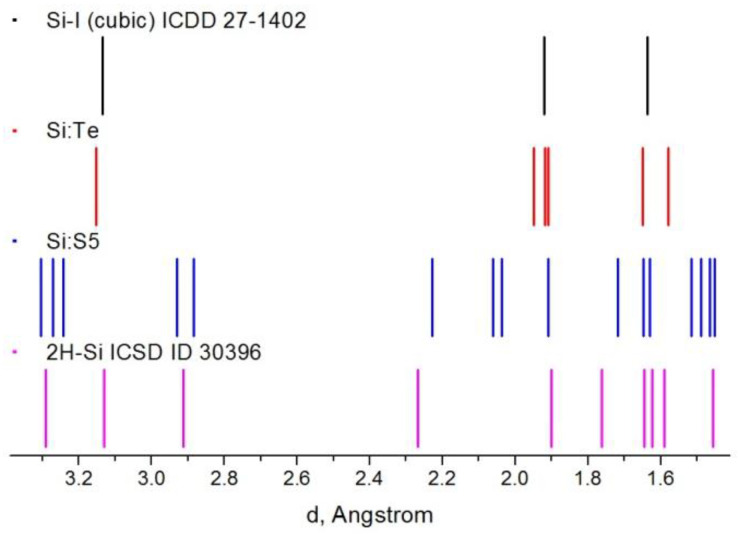
Interplanar distances found in different crystallites of Si:Te (red) and Si:S5 (blue) samples, compared to the literature data for diamond cubic silicon phase (black) and 2H-Si phase (purple).

**Figure 5 materials-15-08842-f005:**
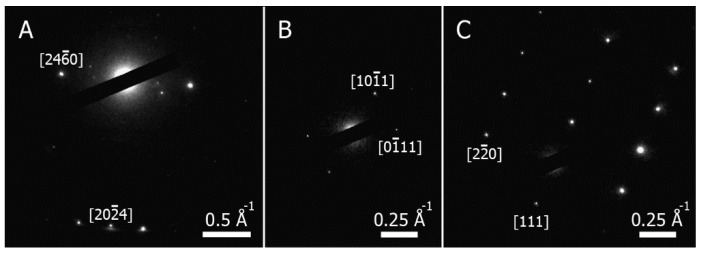
Electron diffraction images of crystallites in: (**A**) Si:S5 sample, indexed according to 2H-Si phase; (**B**) Si:S2 sample, indexed according to 4H-Si, angle between directions equaled 57°; (**C**) Si:Te sample, indexed according to the cubic phase.

**Figure 6 materials-15-08842-f006:**
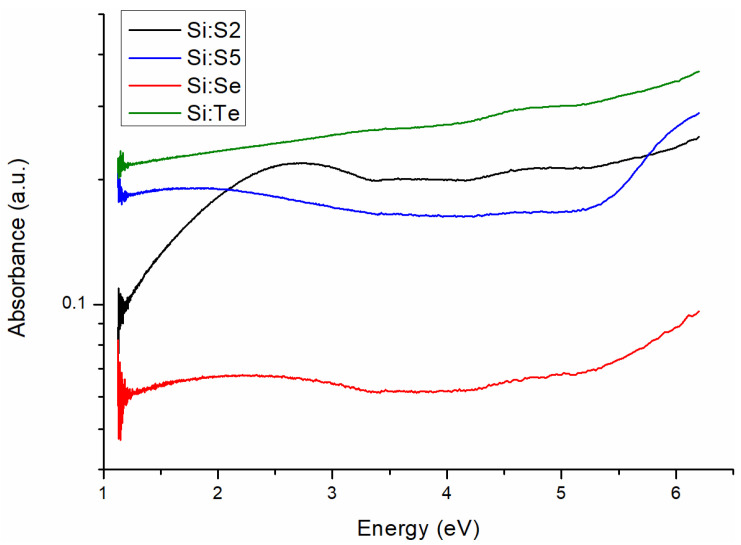
Absorption spectra of samples Si:S2, Si:S5, Si:Se and Si:Te.

**Figure 7 materials-15-08842-f007:**
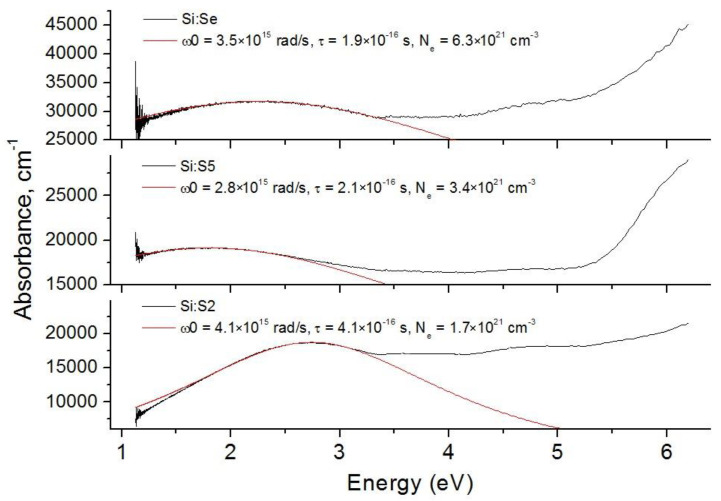
Approximation of absorption spectra and parameters of Drude’s absorption function.

**Figure 8 materials-15-08842-f008:**
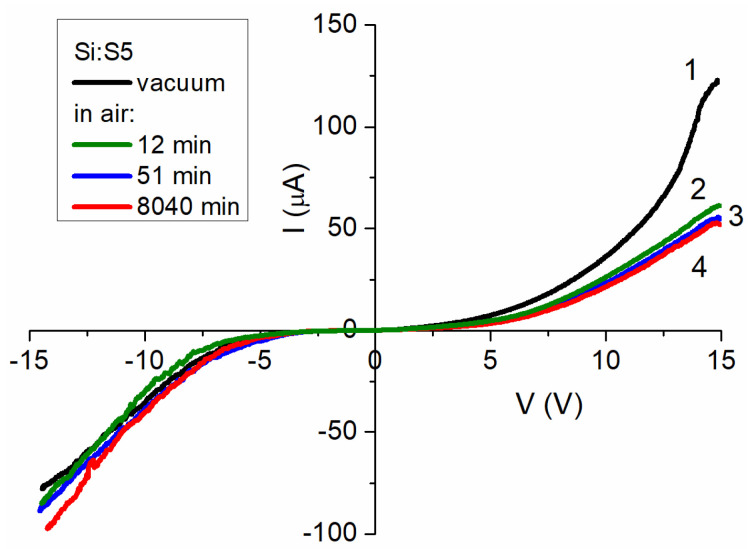
I–V characteristics of Si:S5 film measured: (**1**) at room temperature in a vacuum; (**2**) in air after 12 min of sample contacting the atmosphere; (**3**) in air after 51 min; (**4**) in air after 8040 min.

**Figure 9 materials-15-08842-f009:**
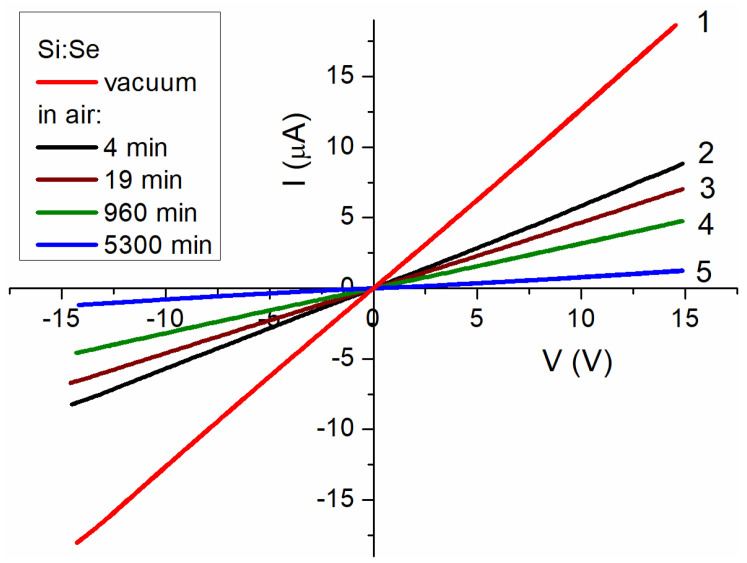
I–V characteristics of Si:Se film measured: (**1**) at room temperature in a vacuum; (**2**) in air after 4 min of sample contacting the atmosphere; (**3**) in air after 19 min; (**4**) in air after 960 min; (**5**) in air after 5300 min.

**Figure 10 materials-15-08842-f010:**
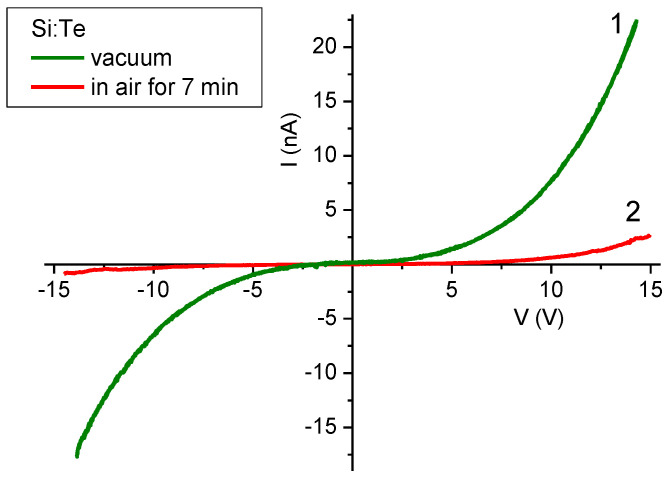
I–V characteristics of Si:Te film measured at room temperature: (**1**) in a vacuum; (**2**) after 7 min of contact with air.

**Table 1 materials-15-08842-t001:** Annealing conditions, impurity content and qualitative frequency of rod-like object observations for samples treated with sulfur.

Sample	Initial. S conc., at.%	T, °C	t, h	Presence of Rods *	S conc., at.% before Etching	S conc., at.% after Etching
Si:S500	10	500	1	-	1.54	0.60
Si:S700	10	700	1	-	1.56	0.50
Si:S900	10	900	1	+	0.94	0.24
Si:S1	20	800	1	-	2.31	0.21
Si:S2	20	825	3	+	1.33	0.12
Si:S3	20	850	1	+	2.28	0.19
Si:S4	20	850	3	++	5.04	0.23
Si:S5	20	850	5	+++	1.90	0.38

* Symbols “+”, “++” and “+++” are used to show the relative amount of rods in corresponding samples, with “+” meaning the lowest and “+++” the highest amount; “-” means that no rods are present in the sample.

**Table 2 materials-15-08842-t002:** Results of annealing in the presence of 20 at.% of S, Se and Te (X stands for chalcogen).

Sample	Morphology	X, at.% before Etching	X, at.% after Etching
Si:S5	Hexahedral rods	1.90	0.38
Si:Se	Faceted crystallites	1.06	0.64
Si:Te	Faceted crystallites and tetrahedral rods	0.60	0.59

**Table 3 materials-15-08842-t003:** Parameters of Drude’s absorption function.

Sample	*ω*_0_, rad/s	τ, s	*N_e_*, cm^−3^
Si:S2	4.1 × 10^15^	4.1 × 10^−16^	1.7 × 10^21^
Si:S5	2.8 × 10^15^	2.1 × 10^−16^	3.4 × 10^21^
Si:Se	3.5 × 10^15^	1.9 × 10^−16^	6.3 × 10^21^

## Data Availability

Not applicable.

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
