# Peer review of "Recrystallization of Si Nanoparticles in Presence of Chalcogens: Improved Electrical and Optical Properties"

_materials, 2022, doi:10.3390/ma15248842_

Round 1

Reviewer 1 Report

Title “Recrystallization of Si in presence of chalcogenes”

The objectives of this work were determining the optimal experimental condition for synthesis of nanosilicon microrods by sulfur doping and studying the effect of heavier sulfur analogs – selenium and tellurium – on structure and electrophysical properties of nc-Si after doping.

Reviewer Comments:

Some explanations in certain points are vague and imprecise and it would be convenient to be clearer and more precise.

Line 54: “can overlap with signals from defects”: recommendation to specify “signals”. Which signals and properties?

Line 85: during the work: during this work

In the section of “Results and discussion”

Figure 1b) and lines between 131-135

The authors mentioned “rare hexagonal objects with diameter of 5 μm are observed” The rare objects need to be identified in the figure, since they hardly can be identified in the figure as authors mention.

The authors referred to “Longer annealing results in increased number of microrods”. The images in figure 1c) obtained after 5h annealing it is not in direction of increasing the number of microrods. Instead, it looks more like some melting process and/or nucleation is involved. It would be helpful if the authors could refer to the figures in more clear and detailed way.

Line 220:  is the resonance angular frequency.

Line 221:  is the plasmon angular frequency (not plasmic )

Line 225: “τ и” symbols for the physics quantities need to be changed.

Line 280: “at low voltages is well described by Ohm’s law and at high voltages” to be clear and accurate the mention of “low” and “high” voltage means would be convenient.

In Conclusion section the authors mention “Annealing with sulfur leads to hexagonal section rods,”: Although it is not completely clear along the explanations, images and results, the section of the rods to be hexagonal. Maybe the authors can clarify more and explained better how to come to this conclusion.

Maybe the authors would like to refer to one of the first published work on this subject: C. Holm and E. Sirtl, “CRYSTALLIZATION OF SILICON FROM THE SILICON—CHALCOGEN VAPOR PHASE”; Journal of Crystal Growth 54 (1981) 253—266

There are some English words wrongly applied and typos, for example.

Line 15: It is not clear the meaning of the word “reflexes”. Recommendation to replace by other suitable word.

Line 17: “respectfully” should be replaced by “respectively”

Line 225: “τ и” symbols for the physics quantities need to be changed.

Some typos like “sulfur ” needs correction  “…sulphur…”

In the title “chalcogenes” needs correction.

Reviewer 2 Report

This article reports the recrystallization and doping of Si nanocrystals with S, Se and Te by annealing in chalcogens’ vapors in vacuum. Various characterizations confirm the structure and morphology of Si particles and their optical and electrical properties. The samples doped with S and Se show high conductivity compared to bare particles. Major revision is recommended. More detailed comments have been listed in below: 

1. The Title should be more specific, including Si nanocrystals and the improved properties. 

2. The results in Section 3.1 should be different from Ref. 30 (Inorganic Chemistry Communications 141 (2022) 109602). Especially, the SEM image (Figure 1c) must not be included, which is the same as that in Ref. 30. 

3. Only XRF, SEM and TEM for doping characterization analysis is not enough. More characterization methods are needed to evaluate the effectiveness of element doping. Such as XRD, EDS, etc.

4. Based on results of S doped Si nanocrystals, the optimal conditions for annealing Si with Se and Te are chosen as 20 at. % of dopant, 850 °Ð¡, 5 h duration. Please add the description. 

Round 2

Reviewer 2 Report

The revised manuscirpt can be accepted now.